# Signal integration at spherical bushy cells enhances representation of temporal structure but limits its range

**Christian Keine[1,2]\*, Rudolf Rübsamen[2], Bernhard Englitz[3]**

[1]Carver College of Medicine, Department of Anatomy and Cell Biology, University of Iowa, Iowa City, United States; [2]Faculty of Bioscience, Pharmacy and Psychology, University of Leipzig, Leipzig, Germany; [3]Donders Center for Neuroscience, Department of Neurophysiology, Radboud University, Nijmegen, Netherlands

**Abstract** Neuronal inhibition is crucial for temporally precise and reproducible signaling in the auditory brainstem. Previously we showed that for various synthetic stimuli, spherical bushy cell (SBC) activity in the Mongolian gerbil is rendered sparser and more reliable by subtractive inhibition (Keine et al., 2016). Here, employing environmental stimuli, we demonstrate that the inhibitory gain control becomes even more effective, keeping stimulated response rates equal to spontaneous ones. However, what are the costs of this modulation? We performed dynamic stimulus reconstructions based on neural population responses for auditory nerve (ANF) input and SBC output to assess the influence of inhibition on acoustic signal representation. Compared to ANFs, reconstructions of natural stimuli based on SBC responses were temporally more precise, but the match between acoustic and represented signal decreased. Hence, for natural sounds, inhibition at SBCs plays an even stronger role in achieving sparse and reproducible neuronal activity, while compromising general signal representation.
DOI: https://doi.org/10.7554/eLife.29639.001

\*For correspondence: christian.keine@gmail.com

**Competing interests:** The authors declare that no competing interests exist.

## Introduction

Acoustically evoked inhibition plays a crucial role in shaping the neuronal activity already at the first central station of the auditory system (*Caspary et al., 1994*; *Kopp-Scheinpflug et al., 2002*; *Dehmel et al., 2010*; *Kuenzel et al., 2011*; *Keine and Rübsamen, 2015*; *Keine et al., 2016*). In a previous study, we demonstrated that inhibition on spherical bushy cells (SBCs) renders their output sparser and more reproducible than their auditory nerve fiber (ANF) input (*Keine et al., 2016*). These transformations persist over a wide range of acoustic stimuli and sound intensities, and can be approximated by an inhibition which we modelled as a scaled subtraction. Functionally, this inhibition emphasizes reliable events and controls the response gain across a wide range of sound levels.

Since in most previous studies, the neuronal activity was recorded either during simple or complex, but synthetic acoustic stimuli, it remains unknown if the inhibitory effect on sound encoding persists also in natural acoustic environments. Here, we extend the range of tested stimuli to natural sounds approximating a gerbil's environment. In such a natural context, the inhibitory influence on the SBC output activity proves even stronger than under complex, but non-natural stimulus conditions tested before. In particular, while for most synthetic stimuli the SBC firing rates generally increase, they remained constant under natural acoustic stimulation.

While transformations of this kind can emphasize certain aspects of the sensory input, they may also deemphasize others. We study this trade-off directly by performing stimulus reconstruction from the population of cells, a technique which has already been successfully applied in cortical

recordings (*Stanley et al., 1999*; *Mesgarani et al., 2009*). In contrast to single-cell analysis, this population-based technique provides an estimate of the overall stimulus information available in a group of neurons. Reconstructions of this type have previously been successfully employed to identify the effect of attention on the neural response (*Mesgarani and Chang, 2012*). We employ the respective reconstructions to analyze the effect of inhibition in the represented stimulus spectrogram.

We find stimulus reconstructions based on the ANF input to be overall more accurate than those from the SBC output. Surprisingly, the share of ANF inputs which are not transmitted to the SBC output (partly blocked by inhibition, *Kuenzel et al., 2011*; *Keine and Rübsamen, 2015*; *Keine et al., 2016*) delivered the most accurate reconstructions, but were temporally more variable. We find that the overall fidelity of representing the stimulus is reduced in SBCs due to the inhibitory gain control, resulting in a lower variance in stimulus reconstruction. Consistent with their role in sound localization, the SBCs' inhibition-shaped output appears to be more focused on restricted, short-term signal representation during variable stimulus conditions, while the overall information about the stimulus is reduced.

## Results

We recorded from spherical bushy cells (SBCs) in the AVCN of anesthetized gerbils in vivo to understand the influence of neuronal inhibition on the encoding of complex acoustic sounds. For this purpose, we presented real environmental sounds (e.g. walking on gravel, singing birds), and performed an explicit population decoding which allows the analysis of the stimulus representation in its original form.

### Modulation of SBC responses under natural acoustic stimulation

To directly investigate the transformation at the ANF-SBC synapse, the neuronal response was separately analyzed for EPSPs which trigger an output spike ($EPSP_{succ}$) and EPSPs which fail to trigger SBC activity ($EPSP_{fail}$) as described previously (*Figure 1A*) (*Keine et al., 2016*). As shown before, these failures of transmission are mostly caused by inhibition at the ANF-SBC junction (*Kuenzel et al., 2011*; *Keine and Rübsamen, 2015*; *Keine et al., 2016*). The acoustic stimulus was composed of seven different segments of varying spectral breadth and featuring rapid amplitude modulations (*Figure 1B+C*). First, we recorded the change in firing activity of ANF input and SBC output for simple pure tones at the units' characteristic frequency. Both ANF input and SBC output rates increased during pure tone stimulation ($ANF_{spont}$ = 71.3 ± 31.8 Hz vs. $ANF_{tone}$ = 234.3 ± 53.9 Hz; $\Delta$ = 163 ± 59.3 Hz, p<0.001; $SBC_{spont}$ = 43.4 ± 18.3 Hz vs. $SBC_{tone}$ = 97.3 ± 33.9 Hz, $\Delta$ = 53.9 ± 27.9 Hz, p<0.001, see *Table 1* for additional details of statistical tests, and *Figure 1—source data 1*, *Figure 1Di*). During natural acoustic stimulation, the ANF input firing rates also increased ($ANF_{spont}$ = 71.3 ± 31.8 Hz vs. $ANF_{natural}$ = 129.7 ± 38.4 Hz, $\Delta$ = 58.4 ± 25.5 Hz, p<0.001), however, contrary to pure tone stimulation, the SBC output activity remained unchanged ($SBC_{spont}$ = 43.4 ± 18.3 Hz vs. $SBC_{natural}$ = 43.1 ± 22.1 Hz, $\Delta$ = 0.3 ± 15.5 Hz, p=0.9, *Figure 1C* for representative trace and *Figure 1Dii* for population data). This effect was persisted when the different stimulus segments were analyzed separately (*Figure 1Diii*). While the increase in SBC firing rates for tones is consistent with previous studies using various synthetic stimuli (*Kopp-Scheinpflug et al., 2002*; *Kuenzel et al., 2011*; *Keine and Rübsamen, 2015*; *Keine et al., 2016*), the constancy for natural stimulation has not been demonstrated before.

The unchanged average firing rates of the SBC output during natural acoustic stimulation was accompanied by an increase in threshold EPSP (Spont = 7.5 ± 2.4 V/s vs. Stim = 9 ± 2.8 V/s, $\Delta$ = 1.4 ± 0.8 V/s, p<0.001, see *Figure 2—source data 1*) and failure fraction (Spont = 0.36 ± 0.2 vs. Stim = 0.65 ± 0.17, $\Delta$ = 0.29 ± 0.1, p<0.001, *Figure 2A*) (see also *Keine et al., 2016*). Together with previous results, this indicates a strong influence of inhibition, which limits the increase in average SBC firing and thereby effectively regulates the SBC output gain. The quality of the neuronal response was assessed by calculating the sparsity (*Figure 2B*) and response reproducibility (*Figure 2C*) for both the complete natural stimulus and the different stimulus segments individually. Consistent with our previous results using synthetic sounds, both sparsity and reproducibility of the

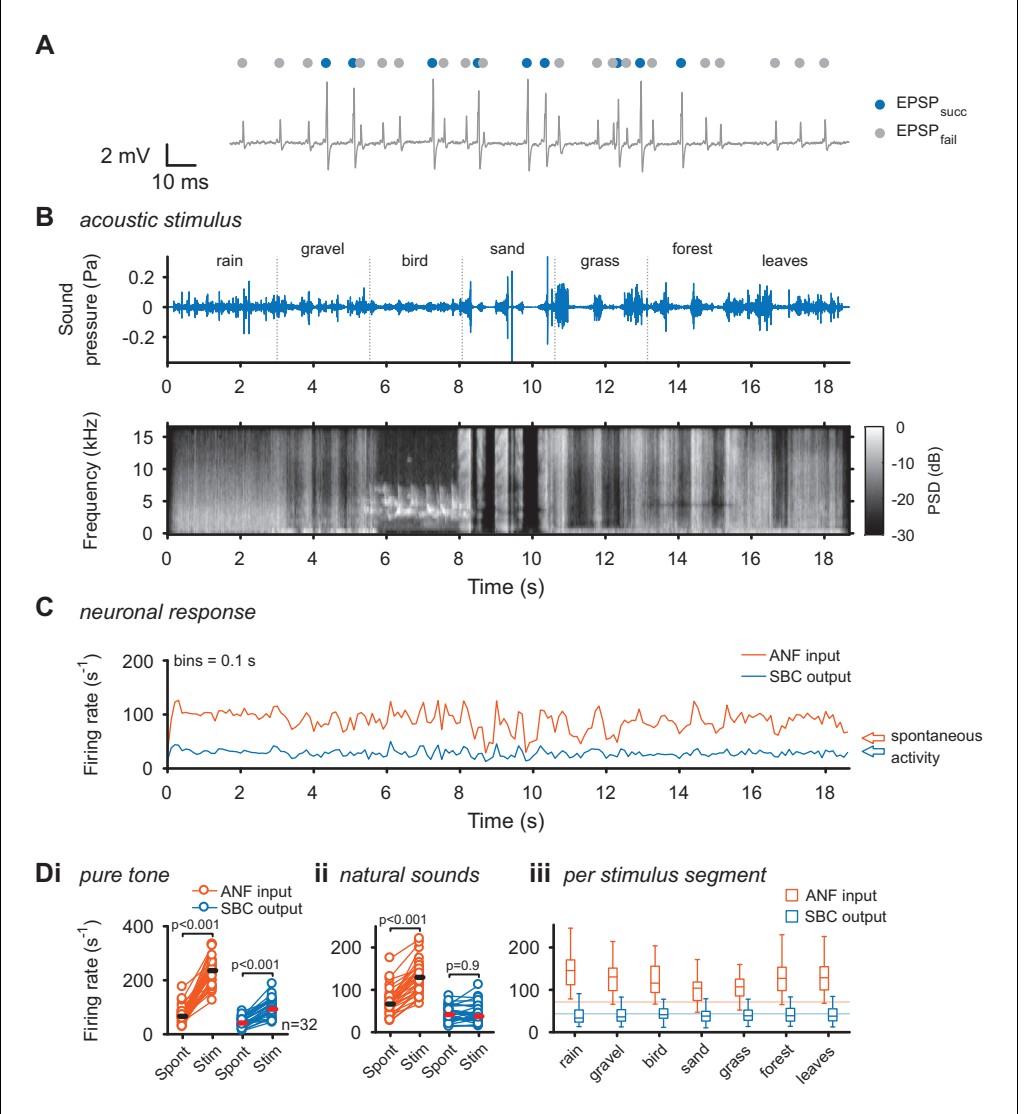

**Figure 1.** Acoustic stimulation with environmental sounds increases auditory nerve firing while leaving the SBC firing rates constant. (**A**) Representative voltage trace of SBC recording during acoustic stimulation. Voltage signals could be divided into EPSP followed by postsynaptic AP (blue dots) and EPSPs which fail to trigger an AP (gray dots). The sum of both types of events comprised the ANF input. (**B**) The acoustic stimulus was composed of a series of environmental sounds (i.e. rain falling, walking on gravel, bird singing, shoveling sand, ripping grass, walking through forest, walking on fallen leaves), containing naturally occurring frequency and amplitude modulations. Top: amplitude profile; bottom: spectrogram. The stimulus was presented at a mean sound pressure level of 40 dB SPL. PSD = power spectral density. (**C**) Firing rates of one representative cell during acoustic stimulation (CF = 2.6 kHz, bin size 0.1 s). Note that the firing rate fluctuations at the ANF level (orange) are higher than at the SBC level (blue). Arrows at the right indicate spontaneous firing rates in the absence of sound. (**D**) Population data on firing rate changes for ANF input and SBC output during pure tone and natural sound stimulation. (i) Pure tone stimulation at the units' characteristic frequency resulted in a firing rate increase in both ANF input and SBC output. (ii) While during stimulation with natural sounds, the average ANF firing increased comparably to pure tone stimulation, the average SBC firing remains at the level of spontaneous activity. (iii) This effect was consistently observed throughout different stimulus segments of the natural sound stimulus. Horizontal lines indicate the average spontaneous firing rate in the absence of sound.

DOI: https://doi.org/10.7554/eLife.29639.002

The following source data is available for figure 1:

**Source data 1.** Firing rates of auditory nerve (ANF) input and Spherical Bushy Cell (SBC) output during spontaneous activity, pure tone and natural acoustic stimulation.
DOI: https://doi.org/10.7554/eLife.29639.003

**Table 1.** Summary of statistical comparisons.

| Source | Parameter | Group 1 (mean ± SD) | Group 2 (mean ± SD) | Test statistics | Statistical test |
|---|---|---|---|---|---|
| *Figure 1* | | | | | |
| Panel Di (pure tones) | Firing Rate ANF | Spont = 71.3 ± 31.8 Hz | Stim = 234.3 ± 53.9 Hz | t(df = 31) = –15.5, p=3.5e-16, U1=0.87 | paired t test |
| | Firing Rate SBC | Spont = 43.4 ± 18.3 | Stim = 97.3 ± 33.9 | t(31) = –10.9, p=3.5e-12, U1=0.59 | paired t test |
| Panel Dii (natural sounds) | Firing Rate ANF | Spont = 71.3 ± 31.8 Hz | Stim = 129.7 ± 38.4 Hz | t(31) = –12.9, p=4.9e-14, U1=0.3 | paired t test |
| | Firing Rate SBC | Spont = 43.4 ± 18.3 | Stim = 43.1 ± 22.1 | t(31) = 0.12, p=0.9, U1=0.03 | paired t test |
| *Figure 2* | | | | | |
| Panel A | Threshold EPSP | Spont = 7.5 ± 2.4 V/s | Stim = 9 ± 2.8 V/s | t(31) = –10.2, p=1.8e-11, U1=0.1 | paired t test |
| Panel B | Failure Fraction | Spont = 0.36 ± 0.2 | Stim = 0.65 ± 0.17 | t(31) = –16.3, p=9.1e-17, U1=0.28 | paired t test |
| Panel Ci | Sparsity | ANF input = 0.09 ± 0.05 | SBC output = 0.17 ± 0.07 | t(31) = –6.9, p=8.2e-8, U1=0.17 | paired t test |
| Panel Di | Reproducibility | ANF input = 0.13 ± 0.09 | SBC output = 0.28 ± 0.18 | Z(N=32) = –4.9, p=8.8e-7, U1=0.2 | Wilcox. sig.-rank |
| *Figure 3* | | | | | |
| Panel K | Corr. Section | ANF = 0.66 ± 0.14 | EPSP = 0.71 ± 0.11 | Z(N=10) = –2.8, p=0.006*, U1=0.1 | Wilcox. sig.-rank |
| | Corr. Section | ANF = 0.66 ± 0.14 | SBC = 0.56 ± 0.16 | Z(N=10) = –2.8, p=0.006*, U1=0.1 | Wilcox. sig.-rank |
| | Corr. Section | EPSP = 0.71 ± 0.11 | SBC = 0.56 ± 0.16 | Z(N=10) = 2.8, p=0.006*, U1=0.35 | Wilcox. sig.-rank |
| | Corr. Time Section | ANF = 0.68 ± 0.14 | EPSP = 0.71 ± 0.14 | Z(N=10) = –2.7, p=0.0117*, U1=0.1 | Wilcox. sig.-rank |
| | Corr. Time Section | ANF = 0.68 ± 0.14 | SBC = 0.58 ± 0.15 | Z(N=10) = 2.7, p=0.0117*, U1=0.15 | Wilcox. sig.-rank |
| | Corr. Time Section | EPSP = 0.71 ± 0.14 | SBC = 0.58 ± 0.15 | Z(N=10) = 2.8, p=0.006*, U1=0.25 | Wilcox. sig.-rank |
| | Corr. Freq. Section | ANF = 0.45 ± 0.042 | EPSP = 0.44 ± 0.071 | Z(N=10) = 1.6, p=0.39*, U1=0.2 | Wilcox. sig.-rank |
| | Corr. Freq. Section | ANF = 0.45 ± 0.042 | SBC = 0.34 ± 0.058 | Z(N=10) = 2.8, p=0.006*, U1=0.85 | Wilcox. sig.-rank |
| | Corr. Freq. Section | EPSP = 0.44 ± 0.071 | SBC = 0.34 ± 0.058 | Z(N=10) = 2.8, p=0.006*, U1=0.5 | Wilcox. sig.-rank |
| | | *Factor 1* | *Factor 2* | | |
| *Figure 4* | | | | | |
| Panel E | Autocorrelation | Height: p=2.9e-25 | Response Type: p=0.002 | | ANOVA, 2-factor |
| Panel F | Spectra | Mod. Rate: p=1.7e-6 | Response Type: p=2.1e-51 | | ANOVA, 2-factor |
| Panel G | Freq. XCorr | Delta Freq: p=1.1e-14 | Response Type p=0.023 | | ANOVA, 2-factor |

*p-values Bonferroni-corrected for N = 3 comparisons.

DOI: https://doi.org/10.7554/eLife.29639.004

SBC output were increased compared to the ANF input (sparsity: ANF input = 0.09 ± 0.05 vs. SBC output = 0.17 ± 0.07, Δ = 0.08 ± 0.06, p<0.001; reproducibility: ANF input = 0.13 ± 0.09 vs. SBC output = 0.28 ± 0.18, Δ = 0.14 ± 0.12, p<0.001). Notably, while the absolute values of sparsity and reproducibility were lower for natural sounds compared to the complex, but synthetic stimuli used previously (*Keine et al., 2016*), the relative increase of both metrics at the ANF-SBC junction was larger during natural acoustic stimulation (sparsity: natural = 105 ± 98% vs. synthetic = 48 ± 49%, p=0.004; reproducibility: natural = 118 ± 74% vs. synthetic = 80 ± 59%, p=0.024, t-test) (*Keine et al., 2016*).

These results corroborate that acoustically evoked inhibition significantly shapes the SBC output activity and results in sparser and more reproducible SBC firing compared to ANF input. Next, we simulated a rate-dependent subtractive inhibition (see Materials and methods and *Figure 2* – supporting *Figure 1* for details) and compared the simulation results to the experimental data. We found that the simulated response (*Figure 2Bii*, green) showed changes similar to the measured SBC output, suggesting that activity-dependent subtractive inhibition is a possible candidate mechanism in shaping SBC output activity during acoustic stimulation.

## Dynamic stimulus reconstruction from population responses

While the gain control at the SBC output seems beneficial for the input to coincidence detector neurons in the MSO, this increase in precision and reproducibility might be achieved at the expense of overall stimulus representation. This is already indicated by the flattened rate-level curves in SBCs

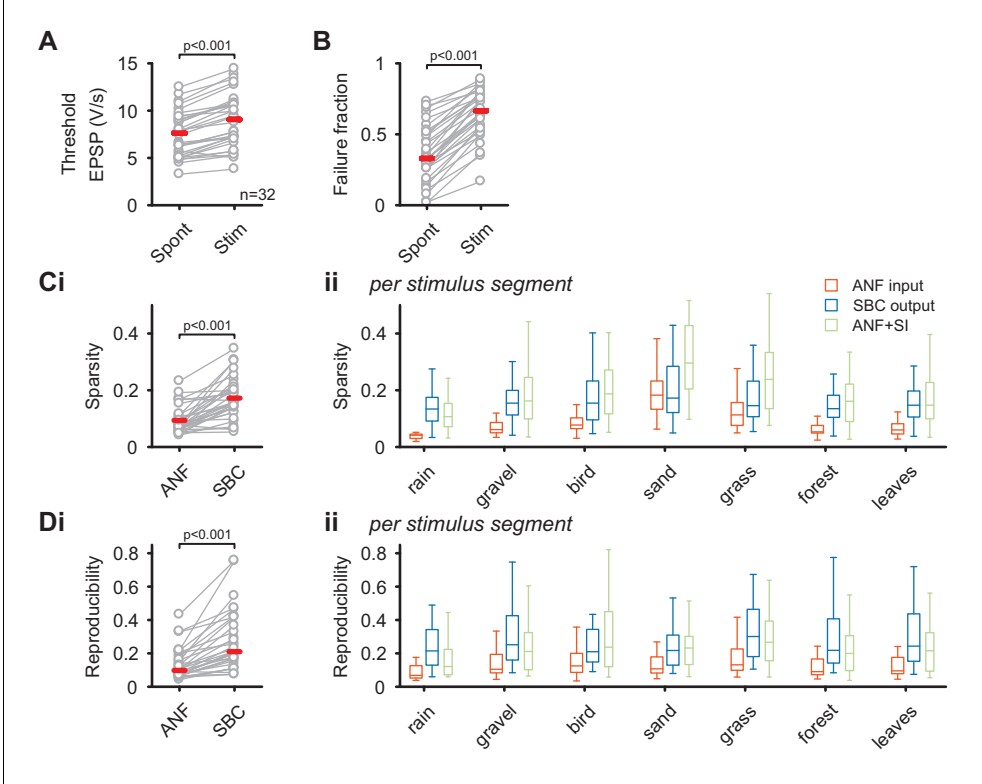

**Figure 2.** SBC output exhibits increased sparsity and reproducibility compared to ANF input which can be attributed to activity-dependent subtractive inhibition. (**A**) During acoustic stimulation the threshold EPSP for AP generation (left) was increased and consequently so was the failure fraction (right), indicating strong inhibition during acoustic stimulation with natural sounds. (**B**) (i) The sparsity of the neuronal response was separately calculated for the ANF input and the SBC output. The SBC output showed consistently higher sparsity than the ANF input. (ii) The increase in sparsity from the ANF input (orange) to the SBC output (blue) was consistently observed for the different stimulus segments. Simulated (subtractive) inhibition (ANF +SI, green) resulted in similar increases in sparsity. Notably, for conditions in which the sparsity of the ANF input was high (i.e. 'sand'), the SBC output did not increase further. (**C**) (i) Similar to sparsity, the reproducibility of the neuronal response increased at the SBC level. (ii) Again, this effect was consistent for the different stimulus segments and well approximated by simulating subtractive inhibition.

DOI: https://doi.org/10.7554/eLife.29639.005

The following source data and figure supplement are available for figure 2:

**Source data 1.** Threshold EPSP (in V/s) and failure fraction during acoustic spontaneous activity and natural acoustic stimulation.

DOI: https://doi.org/10.7554/eLife.29639.007

**Figure supplement 1.** Comparison of different inhibition models in their ability to reproduce the response as well as its sparsity and reproducibility.

DOI: https://doi.org/10.7554/eLife.29639.006

compared to ANFs (*Keine et al., 2016*, *Figure 3*). We therefore estimated how well the neuronal responses of ANFs, SBCs and also the failed EPSPs represent the acoustic stimulus.

Responses of sensory neurons covary with certain aspects of an externally presented stimulus. For single neurons, this covariation is thus often quantified in relation to certain stimulus properties (frequency, sound level, lag, etc.) establishing different types of receptive fields (as in *Keine et al., 2016*). While informative about single-cell properties, these analyses fail to provide a more complete understanding of the representation on the population level, which assesses the different stimulus aspects jointly. For this purpose, it is convenient to combine the population responses and relate them to the original stimulus representation. One general approach of this kind is *stimulus reconstruction* (*Stanley et al., 1999*; *Mesgarani et al., 2009*; *Mesgarani and Chang, 2012*), which

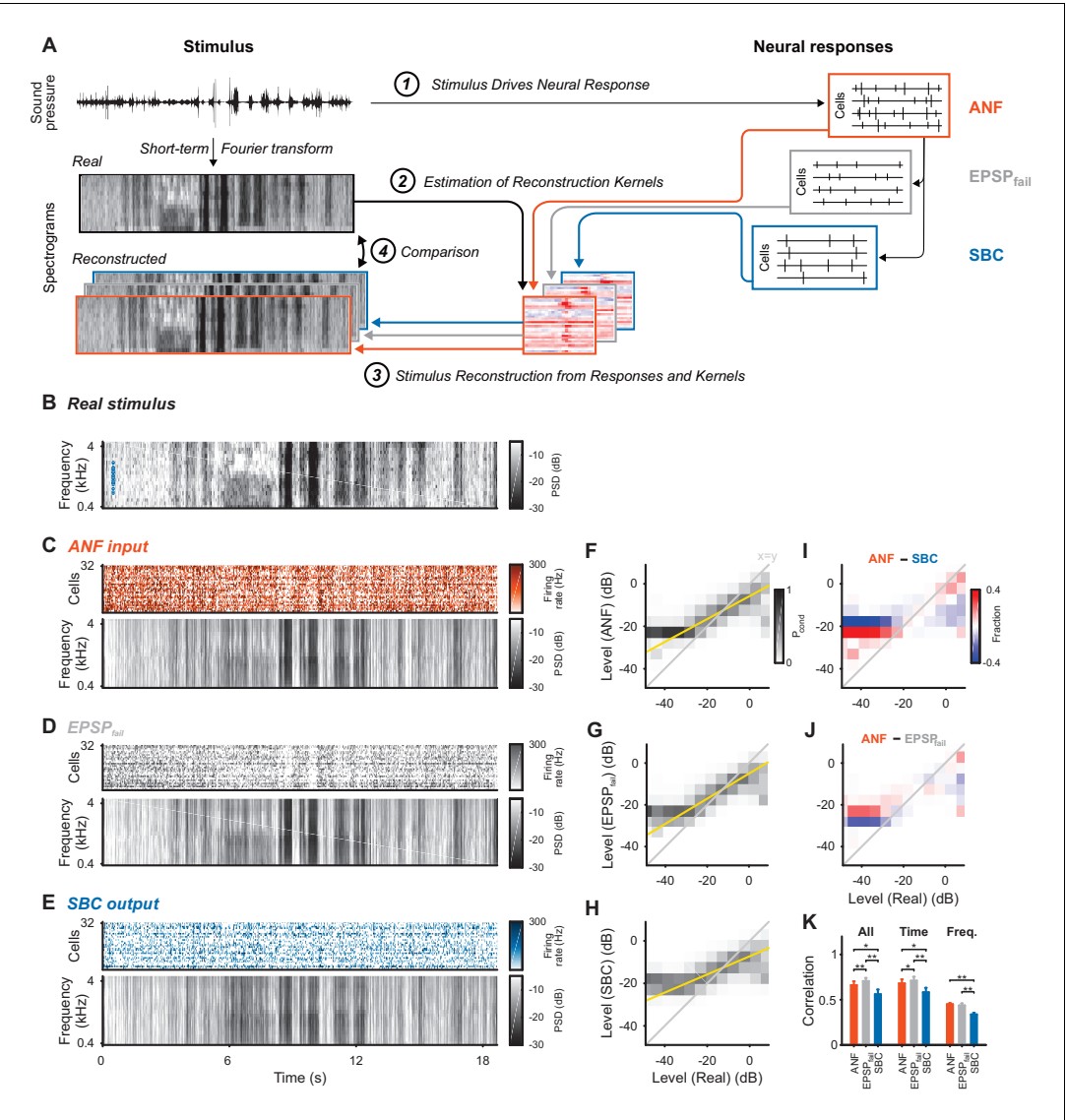

**Figure 3.** Stimulus representation in the SBC responses is less accurate than in the ANF activity or in the inhibited EPSPs (EPSP_fail). (A) Stimulus reconstruction was performed by estimating linear reconstruction kernels (2) for ANF-, EPSP_fail- and SBC responses (1), separately. The respective reconstructions were then used to predict stimulus reconstructions (3), which were then compared to the real stimulus used in the estimation (4). (B) Spectrogram of the real stimulus. The frequency range was reduced from its original range of 16 kHz (see *Figure 1B*) to the range represented in the ANFs' receptive fields (0.3–4.4 kHz, CFs shown as blue dots on the left). Color scales are identical for all spectrograms. PSD = power spectral density. (C) The population response of the ANFs sorted by CF (top) and the ANF-reconstructed stimulus (bottom). The global structure and even the envelope fine-structure is preserved in the reconstruction. For more finely resolved spectrograms see *Figure 4*. (D) The rate of EPSP_fail sorted as above (top) and the EPSP_fail-reconstructed stimulus (bottom). Again, global structure and envelope fine-structure are preserved with inaccuracies in the overall range. (E) The SBC AP population response (top) and the SBC-reconstructed stimulus (bottom). While the envelope fine-structure appears again preserved, the range of the reconstruction is much more limited, i.e. relatively faint parts appear louder in the reconstruction louder than expected (e.g. around 9 s), and vice versa loud parts appear fainter (e.g. after 6 s). (F) The joint histogram across levels between real (abscissa) and reconstructed (ordinate) for ANF responses. The correlation between the two is evident (compare to grey diagonal representing x = y), with a slight deviation below −23 dB, were the reconstructed stimulus did not cover low enough levels (yellow line indicates linear regression). (G) The EPSP_fail-based joint histogram with true level exhibits an overall similar shape as the SBCs (see panel J for detailed comparison). (H) The SBC-based joint histogram is more widely distributed around the diagonal and limited in range (see panel I for detailed comparison). (I) Subtraction of the SBC-based from the ANF-based histogram indicates an increase in width apparent by the negative (blue) margins and the positive (red) spine. (J) Subtracting instead the EPSP_fail-based from the ANF-based histogram, leads to a much smaller difference with an even better correlation around the diagonal for EPSP_fail (blue parts on diagonal) for low and high levels. (K) The correlation between the real and the reconstructed stimulus was significantly worse for SBCs compared with either ANF or EPSP_fail. Correlation was mostly governed by temporal (middle), rather than spectral (right) variations for all three signals (n = 10 cross-validation sections, based on 32 neurons, *p<0.05, **p<0.01, see *Table 1* for exact p-values).

*Figure 3 continued on next page*

*Figure 3 continued*

DOI: https://doi.org/10.7554/eLife.29639.008

The following source data and figure supplement are available for figure 3:

**Source data 1.** Correlation between real acoustic and reconstructed stimulus based on ANF input, SBC output and EPSP$_{fail}$ signals.

DOI: https://doi.org/10.7554/eLife.29639.010

**Figure supplement 1.** For certain postsynaptic response properties, the effect of inhibition appears to be accounted for well by activity-dependent subtractive inhibition (*Figure 2*, and *Keine et al., 2016*).

DOI: https://doi.org/10.7554/eLife.29639.009

performs a prediction of the stimulus based on a multitude of neural responses (see *Figure 3A* for illustration and Materials and methods for details).

Stimulus reconstruction based on the ANF responses (*Figure 3C*) provided a faithful estimate of the original stimulus (*Figure 3B*, reconstructed frequency range was restricted to encompass the cell's receptive fields, see *Figure 4* for zoomed samples), in particular representing large fluctuations in sound level. EPSP$_{fail}$-based reconstructions (*Figure 3D*) appeared similar with even more pronounced representation of sound level, apparent in the population dynamics (top, grey). The SBC-based reconstruction (*Figure 3E*), on the other hand, showed decreased overall representation of

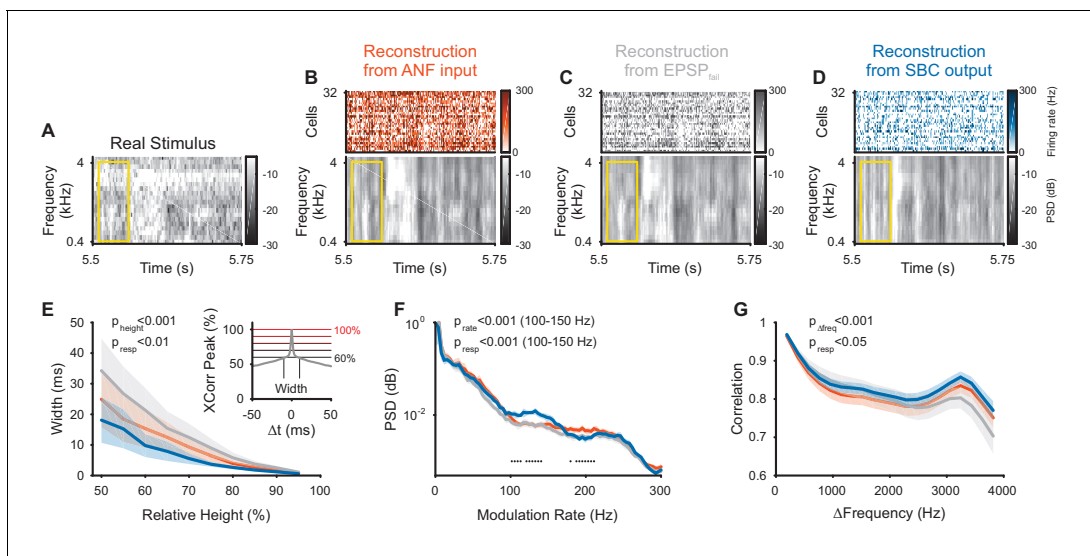

**Figure 4.** Envelope fine-structure properties of the reconstructed stimulus differ between ANF and SBC responses. (**A**) A 250 ms snippet of the real spectrogram (*Figure 3B*) zoomed in. The yellow box indicates a region for comparing the across frequency correlations across the different spectrograms (see G). (**B**) The reconstructed stimulus from the ANFs shows that temporal features of the stimulus can be reconstructed down to approximately 5–10 ms (visual estimate), which is coarser than the resolution of the spectrogram (2 ms). (**C**) The EPSP$_{fail}$ reconstruction appears similar to the ANF reconstruction, with an overall smoother appearance. (**D**) The SBC reconstruction appears more 'vertical', that is, with more correlation across frequencies (see G) and less modulation, but otherwise temporally sharp (see E). (**E**) The temporal precision of the different reconstructions was assessed by the width of the autocorrelation (inset: width of peak), resolved at multiple heights (inset: horizontal lines, black to red) relative to the correlation at $\Delta t = 0$ (inset: maximum). The SBC (blue) reconstruction was most precise, while the EPSP$_{fail}$ (grey) was least precise (2-way ANOVA with factors 'relative height' and 'response type', see panel for p-values, n = 10 stimulus sections). (**F**) The emphasis of the temporal modulations was overall similar with a significant overrepresentation of the 100–160 Hz in the SBC reconstruction (PSD = power spectral density, 2 SEM shown, however, very small variation, see panel for p-values, black dots indicate regions of significant deviation with False-Discovery-Rates at p<0.001, *Benjamini and Hochberg, 1995*). (**G**) The spectral correlation of SBC was larger than for ANFs and EPSP$_{fail}$ for large frequency separations (>2 kHz, see panel for p-values). Correlations were computed for different frequency separations (abscissa), but within each time-bin.

DOI: https://doi.org/10.7554/eLife.29639.011

The following source data is available for figure 4:

**Source data 1.** Properties of the reconstructed stimulus based on ANF, SBC and EPSP$_{fail}$ activity.

DOI: https://doi.org/10.7554/eLife.29639.012

level dynamics with both high and low levels closer to the average stimulus level (average gray level, see Discussion for a mechanistic interpretation).

We compared the dynamic range of the individual reconstructions by computing conditional level densities (CLD, see Methods for details). Each entry is a conditional probability $P_{cond}$ (shown in gray-scale) of a level in the reconstruction (ordinate), given that the same spectrotemporal bin in the real stimulus had a given level (abscissa). The CLDs (*Figure 3F–H*) reflected the differences in level representation (described above) by steeper slopes (yellow lines) for ANFs and EPSP_fail. All three slopes were differed from each other, that is the 95% confidence intervals of the slopes are non-overlapping (ANF: [0.54 ,0.542], EPSP_fail: [0.6 ,0.602], SBC: [0.427, 0.429]). To directly compare the CLDs, we subtracted pairs of CLDs for two response types (*Figure 3I/J*). Differences in level representation were particularly salient at the high and low-level edges (high and low level ends).

The overall reconstruction quality was quantified by cross-validated correlation (*Figure 3K*, Materials and methods) and was lower for SBCs (orange) than for ANFs (blue, p=0.017, Wilcoxon signed rank test, n = 10 stimulus sections, Bonferroni-corrected, same tests and n below, see *Figure 3— source data 1*). Surprisingly, EPSP_fail-based reconstruction quality was better than for ANFs (*Figure 3K*, left, grey, p=0.006) despite a lower number of spikes compared to the ANF input. Temporal dynamics contributed most to the reconstruction quality (*Figure 3K*, middle, p<0.02 for all pairwise comparisons), while the frequency representation was comparably inaccurate (*Figure 3K*, right, p<0.006 for SBC vs. ANF/EPSP_fail, but p=0.39 for EPSP_fail vs ANF). Importantly, both temporal and frequency correlations were decreased for SBC-based reconstructions (p<0.02 for both). This was at least partially caused by the inhibitory gain control, reflected by a reduction in the variance of the reconstructed stimulus (ANF: 21.1 ± 3.3 dB, EPSP_fail: 22.3 ± 4.1 dB, SBC: 15.9 ± 2.0 dB, SBC vs. ANF: p<0.005 and EPSP_fail: p<0.005, EPSP_fail vs. ANF: p=0.32). While the reconstruction quality could potentially be improved with a larger sample size, notably, 32 ANFs lead to a comparable reconstruction quality as >250 neurons in the primary auditory cortex (*Mesgarani et al., 2009*). Lastly, reconstructions from SBC responses simulated as ANF responses subjected to subtractive inhibition (as in *Figure 2*, ANF + SI, see Methods for details and *Figure 2—figure supplement 1*) were statistically comparable to real SBC reconstructions (*Figure 3—figure supplement 1*).

While the SBC-based reconstruction indicates an overall less faithful representation of the acoustic stimulus, they also reflect the envelope fine-structure improvement (*Figure 4A–D*) demonstrated previously in SBC responses (*Dehmel et al., 2010*; *Keine et al., 2016*). The autocorrelation of the SBC reconstruction within a frequency band was sharper compared to ANFs and EPSP_fail (*Figure 4E*, p<0.001 for relative height, p<0.002 for response type, 2-way ANOVA), where the autocorrelation width was assessed at different levels relative to the peak of the autocorrelation (abscissa in *Figure 4E*). This sharpening is probably due to a highlighted frequency range between 100–150 Hz (corresponding to a period of 7–10 ms) in their power-spectrum (*Figure 4F*, p<0.001 for modulation rate, p<0.001 for response type, 2-way ANOVA on the range of 100–150 Hz), which may correspond to the inhibitory time-constant measured in vivo (~10 ms, *Nerlich et al., 2014*; *Keine and Rübsamen, 2015*; *Keine et al., 2016*). Finally, the correlation across frequencies (within a given time bin) was increased for the SBC output compared to the ANF input (*Figure 4G*, p<0.001 for frequency distance, p<0.02 for response type, 2-way ANOVA). We interpret this as a focus on temporal events in any frequency location, with a corresponding loss in representing frequency fine-structure (compare also *Figure 4B/D*).

Taken together, we found that during natural acoustic stimulation, SBC output activity exhibited increased sparsity and reproducibility with SBC firing rates unchanged compared to spontaneous activity. While the SBC output encoded temporal features of the acoustic signal with higher fidelity, the overall stimulus was represented less faithfully compared to the ANF input.

## Discussion

The auditory system faces the challenge to localize and identify sounds in complex acoustic environments. We find that already at an initial stage of the central auditory system, the neural representation of natural sounds is conditioned to be sparser and more reproducible at the SBCs compared to its ANF input. This effect was even larger compared to other complex sounds, consistent with theoretical predictions (*Lewicki, 2002*; *Smith and Lewicki, 2006*). While signal integration at SBCs thus

supports the extraction of temporal features, we found that their ability to represent the stimulus across all stimulus levels is limited in comparison to the ANF input.

The reduction in stimulus representation appears to be largely caused by a compression in the representation, leading to a reduced dynamic range (*Figure 3F–H*, in particular 3I) which missed out on the low and high levels in the stimulus (*Figure 3C–E*). This reduction in represented stimulus range resulted in an overall reduced variance in the SBC stimulus reconstruction. We hypothesize that this limitation in reconstruction gain is a consequence of the inhibitory control on the SBCs' output gain, which flattens their rate-level curves (*Keine et al., 2016*, *Figure 3*). The inhibitory modulation at SBCs appears to specifically remove these extreme levels, as indicated by the improved representation based on the blocked ANF inputs only (*Figure 3D*). Remarkably, the latter representation even improves in comparison with the overall ANF input, indicating that the reduced SBC representation is not due to their lower firing rate in comparison with the ANFs.

All stimulus information available to the auditory system is encoded by the ANF responses and the processing in downstream nuclei will entail emphasizing different subsets of this information. Typically, this has detrimental effects on the non-emphasized part. In the present case, the SBC's focus on temporal information limits the representation of sound level information. The latter is relevant for loudness-based localization (in the azimuth via ILDs, *Galambos et al., 1959*; *Sanes and Rubel, 1988*; *Joris and Yin, 1995*; *Batra et al., 1997*), estimation of elevations using spectral cues introduced via head-related transfer functions (*Blauert, 1997*; *Grothe et al., 2010*), and potentially for sound identification. The ascending input to the LSO (high-frequency SBCs, *Warr, 1966*; *Glendenning et al., 1985*; *Shneiderman and Henkel, 1985*; *Cant and Casseday, 1986*; *Smith et al., 1993*) as well as signal processing in other regions of the cochlear nucleus (e.g. cells in the DCN, *Nelken and Young, 1994*) may emphasize other features of the acoustic stimulus by integrating information differently. For the processing of stimulus envelope, relevant in ILD-based sound localization in the LSO (for review see *Tollin, 2003* and *Grothe et al., 2010*), the optimal processing strategy appears less clear: improving the temporal precision in representing the envelope may be relevant in addition to accurately representing stimulus level.

The increase in sparsity from ANF to SBC for natural stimuli exceeded the increase for synthetic stimuli (*Keine et al., 2016*) and the inhibitory gain control was more prominent for natural stimuli, resulting in SBC firing rates comparable to spontaneous activity. Intrinsic properties of natural acoustic (*Rieke et al., 1995*; *Attias and Schreiner, 1997*; *Nelken et al., 1999*; *Lewicki, 2002*; *Hsu et al., 2004*; *Chechik and Nelken, 2012*, reviewed in *Theunissen and Elie, 2014*) and visual (e.g. *Reinagel and Laughlin, 2001*) stimuli have been highlighted before to provide specific properties of the neural response (typically efficient coding), potentially through evolutionary adaptation. Mechanistically, we think that the differences in spectrotemporal structure between the stimuli cause this effect in multiple ways.

First, the natural stimuli exceed the artificial stimuli in spectral width in all sections. Hence, both the excitatory and the inhibitory inputs represent integration over larger frequency ranges. In addition, since activation across frequencies is less coordinated than in the more local, synthetic stimuli, the ANF is driven more diversely for natural stimuli, and its response is thus less sparse and reproducible for almost all sections (*Figure 2*, compared to Figure 8, *Keine et al., 2016*). Therefore, a similar absolute increase in sparsity, and a smaller absolute increase in reproducibility lead to a larger relative increase in both cases. It remains to be investigated, whether absolute or relative increases are more important for neurons receiving input from SBCs.

Second, the natural stimuli showed a different modulation profile, ranging from little in *rain*, to nearly complete modulation for the segment *sand*. Again, this influences the sparsity and reproducibility already at the level of the ANF input, but will also interact with the inhibitory integration properties. Finally, correlations in the spectrogram may contribute to the differences to synthetic stimuli. Similar findings have been reported for the visual cortex (*Froudarakis et al., 2014*), however, for the auditory brainstem this remains speculation at the current stage.

The integration performed by inhibition could be viewed from the perspective of predictive coding (*Rao and Ballard, 1999*), where the prediction of expected information is subtracted from the current stimulus representation, in order to minimize the number of transmitted spikes for a certain set of statistics, e.g. natural statistics. Based on the progression of time scales represented along the auditory pathway, it could be speculated that the inhibition provides short-term contextual information against which a change in stimulus statistics could be compared. Hence, the integration may

serve to detect low-level changes in the stimulus statistics, similar in principle to the integration of evidence on the cortical level (*Boubenec et al., 2017*).

In summary, the present study using natural stimuli suggests a possible specific adaptation of the inhibitory gain control, leading to unchanged firing rates and larger increases in sparsity and reproducibility compared to synthetic stimuli. These results also indicate, that using acoustic stimuli resembling natural sounds might be necessary to fully understand properties of synaptic integration and signal processing already at initial stages of the auditory system. However, the detailed relation between natural and synthetic stimuli needs to be further explored using hybrid stimuli which isolate specific natural properties and combine them with synthetic stimuli (e.g. SPORCs in *David et al., 2009*).

## Materials and methods

### Animals and surgical procedure

All experiments were performed at the Neurobiology Laboratories of the Faculty of Bioscience, Pharmacy and Psychology of the University of Leipzig (Germany), approved by the Saxonian District Government Leipzig (TVV 06/09) and conducted according to the European Communities Council Directive (86/609/EEC). Animals were housed in the animal facility of the Institute of Biology with 12 hr light/dark cycle and access to food and water ad libitum.

In vivo loose-patch recordings were conducted as described previously (*Keine et al., 2016*). In brief, young adult Mongolian Gerbils aged 6–8 weeks were anesthetized by an intraperitoneal injection of a mixture of ketamine (140 µg/g body weight, Ketamin-Ratiopharm, Ratiopharm, Ulm, Germany) and xylazine hydrochloride (3 µg/g body weight, Rompun, Beyer, Leverkusen, Germany). The animal's skull was exposed and a brass head post glued to the skull to fix the animal in a custom-built stereotactic apparatus in a prone position. Recording electrodes were pulled from borosilicate glass (GB150F-10, Science Products, Hofheim, Germany) to have impedance of 3–5 MΩ when filled with the pipette solution (in mM): 135 NaCl, 5.4 KCl, 1 MgCl$_2$, 1.8 CaCl$_2$, 5 HEPES, pH adjusted to 7.3 with NaOH. The recording electrode was lowered through a hole in the skull into the anterior portion of the ventral cochlear nucleus (AVCN). High-positive pressure was applied (200 mbar) when passing through non-auditory tissue and reduced to 30 mbar when entering the AVCN. When approaching a cell, the pressure was equalized or slight negative pressure (−5 mbar) applied. Single-units were recorded when exhibiting a positive AP amplitude of at least 2 mV and showing the characteristic complex waveform identifying them as large spherical bushy cells of the rostral AVCN (*Pfeiffer, 1966*; *Winter and Palmer, 1990*; *Englitz et al., 2009*; *Typlt et al., 2010*).

### Acoustic stimulation

Recordings were performed in a sound-attenuating and electrically isolated chamber (Type 400, Industrial Acoustics, Niederkrüchten, Germany). Acoustic stimuli were generated by custom-written Matlab (RRID:SCR_001622) functions and delivered via a custom-built earphone (DT48, Beyerdynamic, Heilbronn, Germany) positioned just in front of the ear canal. Acoustic stimuli were composed of environmental sounds and consisted of seven segments of length 3.46 s with $\cos^2$ amplitude transitions of 460 ms between consecutive segments to prevent unexpected transients (see *Supplementary file 1* for the audio file containing the stimulus). The stimulus had a total length of 18.7 s and was presented at least 20 times for each cell at 40 dB SPL with maximal sound intensities of 85 dB SPL.

### Data analysis

Recorded voltage signals were digitized at 97.7 kHz (24 bit, RP2.1, Tucker-Davis Technologies) and filtered between 5 Hz and 7.5 kHz using a zero-phase digital IIR filter. Neuronal signals were detected by the fast upward stroke of the excitatory postsynaptic potential (EPSP) and separated into events which successfully trigger a postsynaptic AP (EPSP$_{succ}$) and events that fail to trigger a postsynaptic action potential (EPSP$_{fail}$). Sparsity and reproducibility of the neuronal response were separately computed for the ANF input (EPSP$_{succ}$ + EPSP$_{fail}$) and the SBC output (EPSP$_{succ}$).

The *sparsity* of the response rate was computed as the variance-based method described previously (*Rolls and Tovee, 1995*; *Willmore and Tolhurst, 2001*) with sparsity defined as

$$S = 1 - \langle r(t) \rangle_t^2 / \left\langle r(t)^2 \right\rangle_t$$

where $\langle . \rangle_t$ indicates an average over time. Its values range between 0 (maximally dense) and 1 (maximally sparse).

The *reproducibility* of the neural response was computed in two steps. First, the raw cross-correlation was computed between two trials, divided by ($S_1 S_2/N_{bins}$), that is, the product of the number of spikes in each trial, divided by the number of time bins. Second, the cross-correlation was averaged across all pairs of non-identical trials, and then normalized by the number of such pairs, equal to N (N-1)/2, where N is the total number of trials. Finally, the cross-correlation at time 0 was chosen, and '1' subtracted from it, in order to obtain a measure which equals 0 for a random process with fixed rate (aside from the centering around 0, it is thus very similar to the correlation index by *Joris et al., 2006*). Values > 0 indicate above chance correlation in the response between trials, beyond what would be expected based on correlation in rate alone. Importantly, reproducibility can attain values >1, with an upper limit determined by the firing rates in the different trials.

## Stimulus reconstruction

The set of responses from all recorded neurons was used to re-estimate the stimulus spectrogram. A linear reconstruction approach was implemented, which amounts to a linear regression between the neural responses as predictors, and the spectrogram of the stimulus, computed as the absolute value of the short-term Fourier transform, as dependent variables (*Mesgarani et al., 2009*; *Mesgarani and Chang, 2012*). This kind of population-based approach allows a combination of the overall information available in the neural response for accurately re-estimating the stimulus. Explicit stimulus reconstruction provides a standardized way to compare stimulus representations along stages of a sensory system, here between the ANF, SBC, and failed EPSPs. Clearly, the brain may use different strategies internally to decode or transform this information, although the present approach has proven useful for approximating the resulting percept (*Mesgarani et al., 2009*; *Mesgarani and Chang, 2012*).

For this purpose, both the response and the stimulus spectrogram were resampled at 2 kHz. For the stimulus this was achieved by allowing neighboring time-bins to overlap by multiple samples, concretely the stimulus was divided into overlapping sections of 512 samples, starting at round ($i_t \times SR_{sound}/SR_{spectrogram}$) for each time step $i_t$, the acoustic stimulus' sampling rate $SR_{sound}$ = 97.65625 kHz, and the desired spectrogram sampling rate of 2 kHz. Neighboring stimulus bins are thus not independent, but the match between stimulus and response sampling rate is required for the general estimation procedure. Based on the chosen stimulus representation, no phase-based fine-structure predictions are possible, therefore, all analyses relating to temporal precision thus relate to *envelope fine-structure*. All displayed spectrograms are shown in the same scale, dB scaled (10 log10).

The reconstruction procedure was carried out as described in detail before (*Mesgarani et al., 2009*). Briefly, with the stimulus spectrogram denoted by S(t,f), the responses by R(t,n), and the reconstruction kernels per frequency band by $g_f(\tau, n)$, the assumed response-stimulus relation can be written as

$$\hat{S}(t,f) = \tau \sum \sum_n g_f(\tau,n) R(t-\tau,n) \tag{1}$$

The kernels $g_f$ can be estimated via the classical normal equation

$$g_f = \left( C_{RR}^{-1} + \lambda\, I \right) C_{RS_f} \tag{2}$$

with the cross-correlation term $C_{RS_f} = RS_f^T$, and the response correlation matrix $C_{RR} = RR^T$, including a ridge regression term to avoid degeneracies in the inversion of $C_{RR}$ (see *Mesgarani et al., 2009* for a more detailed derivation) with $\lambda = 0.1$. During crossvalidation, *Equation 1* is then used to reconstruct the stimulus from estimates of $g_f$ on the non-predicted stimulus sections. Each $g_f$ is

based on the activity of all neurons $n$ at a range of delays $\tau$, and thus given by a matrix (see *Figure 3A* middle bottom for an example). The stimulus was constrained to 300–4400 Hz, slightly extending the range of the characteristic frequencies (CFs, [1.18, 3.05] kHz) of the present sample (see *Figure 3B*, blue dots indicate the CFs of all cells).

The quality of reconstruction was assessed using correlation coefficients between the original and the reconstructed stimulus. The displayed values (*Figure 3*) are averaged between the cross-validated (divided into 10 segments) and the *insample* estimates, based on considerations regarding the influence of noise on estimating model-performance (*Ahrens et al., 2008*). Further, the original stimulus and the reconstruction were compared using conditional level densities (see below, *Figure 3*), the width of the autocorrelation (*Figure 4E*), the frequency spectrum of the activations in the spectrogram (*Figure 4F*), and the spectral auto-correlation (*Figure 4G*).

The conditional level densities (CLD) were computed as the joint histogram across levels between the real and the reconstructed spectrogram, normalized by the probability of the level in the real stimulus. The normalization was introduced to highlight the differences that occur mostly at the stimulus extremes, which are less probable in the distribution of levels. Hence, a column in a CLD sums to one, and constitutes the (empirical) conditional probability density of the estimated spectrogram, relative to the original spectrogram. We also computed differences of two CLDs to highlight changes between reconstructions (e.g. *Figure 4I,J*): the difference here indicates a relative (i.e. between the reconstructions) prevalence/scarcity of levels in the reconstruction in relation to the real stimulus.

The spectral autocorrelation was computed as a cross-correlation across frequencies at every timestep, averaged over the length of the stimulus (*Figure 4G*). It indicates how predictable the envelope level is across frequencies for the different reconstructions.

## Simulation of subtractive inhibition

We extended our simulation of subtractive inhibition to include an estimate of previous activity in order to match it more closely with the experimentally measured delay and temporal dynamics. Inhibition was modelled to depend on the recent history of ANF firing activity in this CF range: First the PSTH of a given ANF input was time shifted by a time $t_c$, and then integrated with an exponential kernel with a time-constant $\tau_I$ to obtain an intermediate representation AI(t). The resulting signal was passed through a static, sigmoidal nonlinearity given by

$$SI(t) = \frac{I_0}{1 + e^{-S\,(AI(t) - O)}}$$

The sigmoid's shape is controlled by $I_0$, S and O, where $I_0$ is the instantaneous rate of inhibition that is maximally subtracted, S is the (inverse) slope of activation, and O the value of the integrated signal AI, where the sigmoid reached 50% of $I_0$, hence, controlling the horizontal offset. These three steps account for the integration ($\tau_I$), nonlinear spike-elicitation ($S$, $I_0$, $O$) and the conduction delay ($t_c$). We manually estimated a set of parameters that accounted best for each SBC's failure rates under spontaneous and natural stimulation, as well as the overall reconstruction quality and variance (see *Figure 2—figure supplement 1* and *Figure 3—figure supplement 1*): The first three parameters were set fixed across all cells to $t_c = 0.5\,ms$, $\tau_I = 3\,ms$, $S = 5$. The last two parameters were set individually per cell to match the experimentally observed failures rates (*Figure 2—figure supplement 1*). Automatic fitting was infeasible since there appeared to be many plateaus due to the discrete nature of the spikes.

The resulting output of the model constitutes a rate of spikes and was then subtracted from the neurons instantaneous activity by deleting the corresponding number of spikes (randomly across trials) in this time bin. If the number of existing spikes per bin was lower than the number of spikes allocated to subtraction, the individual bin was set to 0.

Prior to applying the subtractive inhibition, the spontaneous failure rate was accounted for by removing a random set of spikes, whose size was matched to the spontaneous failure rate of the cell.

## Statistics

Data sets were tested for Gaussianity using the Shapiro-Wilk test (*Shapiro and Wilk, 1965*). Within-subject comparisons were performed by paired *t*-test (normally distributed) or Wilcoxon signed rank test (otherwise). For interpretation of all results, a p-value less than 0.05 was deemed significant, where p-values$<10^{-5}$ are reported as p<0.001 in the text and figures. The effect size was calculated using the MES toolbox in Matlab (*Hentschke and Stüttgen, 2011*) and reported as Cohen's U1. No statistical methods were used to pre-determine sample size. Exact p-values and test statistics are summarized in *Table 1*.

## Acknowledgements

This work was supported by the German Research Foundation (DFG Priority Program 1608 'Ultrafast and temporally precise information processing: Normal and dysfunctional hearing' [RU390/19–1, RU390/20–1]), and Marie Sklodowska Curie Fellowship 660328. The authors thank the three anonymous reviewers for their constructive feedback which substantially improved the manuscript. The authors declare no competing financial interests.

## Additional information

### Funding

| Funder | Grant reference number | Author |
|---|---|---|
| Deutsche Forschungsgemeinschaft | RU 390/19-1 | Rudolf Rübsamen |
| Deutsche Forschungsgemeinschaft | RU 390/20-1 | Rudolf Rübsamen |
| European Commission | Marie Sklodowska Curie Fellowship 660328 | Bernhard Englitz |

The funders had no role in study design, data collection and interpretation, or the decision to submit the work for publication.

### Author contributions

Christian Keine, Conceptualization, Data curation, Software, Formal analysis, Validation, Investigation, Visualization, Methodology, Writing—original draft, Writing—review and editing; Rudolf Rübsamen, Conceptualization, Resources, Supervision, Funding acquisition, Writing—review and editing; Bernhard Englitz, Conceptualization, Software, Formal analysis, Validation, Methodology, Writing—original draft, Writing—review and editing

### Author ORCIDs

Christian Keine ![ORCID] http://orcid.org/0000-0002-8953-2593
Bernhard Englitz ![ORCID] http://orcid.org/0000-0001-9106-0356

### Ethics

Animal experimentation: Animal experimentation: All experiments were approved by the Saxonian District Government, Leipzig (TVV 06/09), and conducted according to the European Communities Council Directive (86/609/ EEC).

### Decision letter and Author response

Decision letter https://doi.org/10.7554/eLife.29639.020
Author response https://doi.org/10.7554/eLife.29639.021

## Additional files

**Supplementary files**
• Source code 1. Calculation of threshold EPSP.
DOI: https://doi.org/10.7554/eLife.29639.013

• Source code 2. Calculation of reproducibility.
DOI: https://doi.org/10.7554/eLife.29639.014

• Source code 3. Calculation of sparsity.
DOI: https://doi.org/10.7554/eLife.29639.015

• Source code 4. Simulation of activity-dependent subtractive inhibition.
DOI: https://doi.org/10.7554/eLife.29639.016

• Supplementary file 1. Related to *Figure 1*. Waveform audio file of presented natural stimulus. The stimulus was presented monaurally at a sampling rate of 97.65625 kHz.
DOI: https://doi.org/10.7554/eLife.29639.017

• Transparent reporting form
DOI: https://doi.org/10.7554/eLife.29639.018

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
