## [Decision Letter]

Thank you for submitting your article "Signal representation is degraded by temporal sharpening through inhibition in the auditory brainstem" for consideration by *eLife*. Your article has been reviewed by three peer reviewers, and the evaluation has been overseen by a Reviewing Editor and Andrew King as the Senior Editor. The reviewers have opted to remain anonymous.

The reviewers have discussed the reviews with one another and the Reviewing Editor has drafted this decision to help you prepare a revised submission.

Summary:

This communication follows up on a previous report (Kiene et al., 2016) regarding the "sparsity" of information represented in spike trains in the bushy cells of the gerbil cochlear nucleus. The previous report demonstrated that inhibition was part of the mechanism involved in improving the temporal representation of certain sound features using amplitude and frequency modulated tones, and using randomized γ tone sequences; one of the observations with these experiments was that the output firing rate of the bushy cells was significantly lower than the firing rate of the inputs (as defined by pre-potential "spikes"). The previous study also focused on determination of synaptic mechanisms in structuring the spiking responses of the bushy cells. The present study extends this by using a set of "natural" environmental stimuli that have varying spectral and temporal structure, although most (with the exception of the bird song) appear as slowly modulated wideband noises with varying slow spectral slopes. The results are consistent with, and build on, the previous study, and are an interesting contribution to the literature, given that naturalistic sounds or their lab-constructed synthetic components, have been relatively sparsely studied in the auditory brainstem nuclei (primarily as human speech sounds and various reduced versions of speech), and may provide important clues as to how sensory perceptions are constructed from the somewhat deconstructive processes of the auditory periphery.

There are two really interesting results in this communication. First, the average firing rate of spherical bushy cells (SBCs) is not increased during stimulation with these sound sets, whereas the average rate in the auditory nerve fibers (deduced from the pre-potential) does increase, and this was fairly independent of the stimulus category. This is remarkably different from the responses to narrow band (tonal) stimuli, and to unmodulated wide band noise stimuli. The increase in the sparsity and reproducibility of spiking in the SBCs confirms what should be predicted from the previous study, but provides generality by extending it to a wider class of stimuli, and it is very interesting that the reproducibility and sparsity measures are higher for the natural stimuli than the previous synthetic stimuli. The second interesting result is that reconstruction of the spectro-temporal pattern of the stimulus is somewhat degraded in the SBCs, although some (but maybe not all?) of the temporal aspects are "enhanced".

Essential revisions:

1) The terms "sparsity" and "response reproducibility" should be defined clearly, so they can stand independently of the earlier paper. A reader would benefit from some context on these numbers to know what is a "good" or "bad" value.

2) Value-laden terms such as "improvements" (Results section) should be avoided. One might benefit from more nuance than is conveyed by terms such as "reduced" fidelity or accuracy, and "impoverished" correlations (Abstract, Introduction and Results section). A change in information from auditory nerve to bushy cell is expected, and could be considered selection, rather than degradation, as the title suggests. The consequence of this is not well captured in the Discussion section. There is a lot more to the auditory system than just bushy cells, after all, so hearing should be just fine if one cell type disregards a certain type of information in favor of another.

3) It would be useful to know the responses of other unit types in these studies. It would also be useful to know if the authors found any primary-like units without a prepotential. Did all cells show spike failures? How does this finding fit with other studies on primary-like units which have used both simple and complex sounds (e.g. AM, FM, steady-state vowels) and found increases in response rate? Did you record the responses to single tones to construct a PSTH and/or receptive field? If so, did the units discharge above their spontaneous rate? What did the receptive field look like? What was the distribution of BFs in these units? How should we think about specificity in the output of SBCs vs. generality of the inhibitory input?

4) Subtractive inhibition (Figure 2). It seems the goal here is to show how inhibition enhances timing precision among bushy cells with respect to auditory nerve fibers. This is an important idea and should be fleshed out more, where alternatives are clearly delineated and tested. The hypothesis seems to be that temporal precision is enhanced when inhibition tracks activity with some delay. This is reasonable based on the measured EPSP(fail) rates. How these translate into the parameters for the inhibitory model should be clearly depicted. But additional analyses are needed to conclude that the pattern of inhibition is responsible for enhanced temporal precision. For example, how is temporal precision influenced by uniformly deleted spikes (tonic inhibition), randomly deleted spikes, using an opposite pattern of spike deletion (heavier following sparse activity), activity-dependent decay of different depths (I(0)? S?) and timecourses (tau1)? Also, to make the approach clear, it would be helpful to show examples of original spike train, modeled inhibition, modified spike train, and sparsity/reproducibility before and after such manipulation. Please plot on a timescale that shows the differences to best advantage. What is the variance referred to in the Results section?

5) Population vs. paired analysis. The approach taken here was to use the entire population of auditory nerve activity and the entire population of bushy cell activity to try to reconstruct the sound signal. Please explain the reasoning behind this approach. It seems to introduce some problems, the most obvious being frequency sensitivity of individual units. That is, one does not expect a single cell to be able to represent much of a sound, and a correction for this was confusingly mentioned in passing (subsection “Stimulus reconstruction”). The critical issue is how completely the individual bushy cell represents the stimulus compared to its inputs. Doing this comparison cell by cell would be more meaningful.

6) The temporal enhancement would be expected based on the mechanisms (potassium conductances, inhibition), following observations by Joris et al. (and others, including work by the present authors) on the improvements in temporal representation of simple sounds in the bushy cell pathways. However, here the sharpened representation of temporal features of the stimulus likely does not derive from the ability of bushy cells to carry the timing of stimulus "fine structure", but it appears to be in carrying timing of envelope fluctuations, at least at the scale of analysis used here. Although this is interesting, it should be clarified, as the term "fine-structure" as used in the manuscript is not always consistent with its use in other parts of the literature, where it refers more closely to the instantaneous waveform than the waveform over a several msec period.

7) The discussion should touch more carefully on the improvement in sparsity and reproducibility for the natural stimuli compared to the synthetic stimuli used in the previous study, both in terms of potential mechanisms and the importance (if there is one) of the representation. How do the authors think this change came about, and why or how is it important?

8) The paper would benefit from discussion of other studies that have considered natural vs. artificial stimuli (e.g. Rieke, Bodnar and Bialek, (1995), Reinagel, (2001)). Finally, you should address how these observations relate to localization (or not) or other aspects of parsing and/or reconstructing the acoustic environment. Some concepts related to the framework of recent work from the last author's group (Boubenec et al., 2017) might present an interesting starting point.

---

## [Author Response]

Essential revisions:1) The terms "sparsity" and "response reproducibility" should be defined clearly, so they can stand independently of the earlier paper. A reader would benefit from some context on these numbers to know what is a "good" or "bad" value.

We have added the full definitions now in the Methods section of the revised manuscript, as well as providing an aid for interpretation there. In this process, we have noted that in the original publication the normalization was not fully spelled out, which may lead to some confusion. We propose to amend the definition there as well to align the descriptions and avoid potential confusion.

2) Value-laden terms such as "improvements" (Results section) should be avoided. One might benefit from more nuance than is conveyed by terms such as "reduced" fidelity or accuracy, and "impoverished" correlations (Abstract, Introduction and Results section). A change in information from auditory nerve to bushy cell is expected, and could be considered selection, rather than degradation, as the title suggests. The consequence of this is not well captured in the Discussion section. There is a lot more to the auditory system than just bushy cells, after all, so hearing should be just fine if one cell type disregards a certain type of information in favor of another.

We think the – by now typical – attempt to squeeze too much information into a short title has created an impression that was not intended: we fully agree with the reviewer that this selection of information at the SBC junction will not be mirrored by most cells in the auditory brainstem. Other cells will likely select for different information, in line with the processing needs of their subsystem. Quite conversely, we in fact wanted to make exactly this point (both in the manuscript as a whole and the Discussion section), that given the 'task' of the SBCs in the auditory system, they appear to be selective for temporal information, and on the flipside, miss out on some sound level-information. While we agree that this conclusion is not entirely unexpected (given a limited amount of coding capacity available for a single neuron), however, the stimulus reconstruction analysis makes this notion explicit.

Another level of quantification could for example be an information theoretic account that tries to specifically account for level and timing information and explicitly analyses the degree to which these are represented in the response. In accordance with the reviewer’s comment we emphasize more strongly in many places that a) this result is (at this point) only established for the SBC (in particular in the Discussion section, but also the Introduction), and b) reduced/removed the value-aspects from the terminology. 'Improvements' are now only reported for the stimulus reconstructions, where the quality in fact varies between 0 (poor) and 1 (excellent).

3) It would be useful to know the responses of other unit types in these studies.

The present study specifically focused on the ANF-to-SBC synapse and the integration of excitation and inhibition on processing of natural sounds. We therefore did not record from other unit types in the CN. It is difficult to predict what their response would be like, since not many studies have compared synthetic and natural stimuli in this respect. From the present study, we can at least conclude that ANFs show an increase in firing rates to both tonal and natural stimuli.

It would also be useful to know if the authors found any primary-like units without a prepotential. Did all cells show spike failures?

Units in this study were exclusively recorded from the rostral AVCN, the location of large, low-frequency spherical bushy cells (Bazwinsky et al., 2008) in line with previous studies investigating the integration of excitation with acoustically evoked inhibition in SBCs (Kuenzel et al., 2011; Keine and Rübsamen, 2015; Keine et al., 2016).

During data collection, we did not encounter primary-like units without a prepotential, although in some units the prepotential was very small once the loose-patch configuration was established (see also the first point of the figure comments for further clarification of this issue). However, the complex waveform and the prominent EPSP are characteristic for SBCs that receive large endbulb inputs and markedly different compared to the biphasic waveform of other cell types in the CN, e.g. stellate cells.

All recorded units showed spike failures, albeit to different degrees, ranging from 2% to 73% in the present sample, consistent with previous observations (Kuenzel et al., 2011; Keine and Rübsamen, 2015; Keine et al., 2016). All units exhibited increased failure fractions during sound stimulation, regardless of their spontaneous failure fraction (see Author response image 1 below). As expected, units with high spontaneous failure rates showed lower increases in failures during stimulation, as a higher number of spontaneous failures will leave fewer transmission events that can be suppressed during acoustic stimulation. Hence, all recorded SBCs were qualitatively of a similar phenotype with some quantitative differences between them.

**Author response image 1. respfig1:** Relation between spontaneous and driven failures. (A) Acoustically driven failure rates always exceeded spontaneous failures, but correlated well with each other. This suggests that the effects of spontaneous failures and stimulus-driven failures (by inhibition or otherwise) are 'additive', although we cannot discern, whether this is true addition (i.e. FF(s + d) = FF(s) + FF(d)) or just a 'greater than' relation (i.e. FF(s+d) > FF(s)). (**B**) The additional failures during stimulation (i.e. Driven – Spontaneous) showed a slight negative correlation. We think this dependence is expected, since in units with higher spontaneous failure fractions, only a smaller fraction of successful transmission events remains, that can still fail during stimulation.

How does this finding fit with other studies on primary-like units which have used both simple and complex sounds (e.g. AM, FM, steady-state vowels) and found increases in response rate? Did you record the responses to single tones to construct a PSTH and/or receptive field? If so, did the units discharge above their spontaneous rate? What did the receptive field look like? What was the distribution of BFs in these units?

Yes, for every unit in this study, a pure tone tuning curve was recorded to estimate the units’ frequency response area. The response areas were similar to the ones reported in previous studies (both in our studies Keine and Rübsamen, 2015; Keine et al., 2016, and others before, e.g. Winter and Palmer, 1990; Caspary et al., 1994; Kopp-Scheinpflug et al., 2002; Dehmel et al., 2010; Kuenzel et al., 2011; Typlt et al., 2012). Given the short format of the Research Advance, and the idea that it builds upon a previous publication, we did not describe these properties again here. The BFs of the units are shown in Figure 3 as blue dots, and the receptive fields was V-shaped with high-frequency inhibitory sidebands, with inhibition found throughout the receptive field (see Keine et al., 2016 for examples and averages). Thus, the sound-driven firing rates at BF stimulation were also increased, resulting in firing rates above the spontaneous rate. This is in accordance with results from previous studies: For tonal stimuli, SBCs show a reduced response gain (Kopp-Scheinpflug et al., 2002; Kuenzel et al., 2011, 2015; Keine and Rübsamen, 2015; Keine et al., 2016) and partly non-monotonic rate level functions (Winter and Palmer, 1990; Kopp-Scheinpflug et al., 2002; Kuenzel et al., 2011, 2015; Keine and Rübsamen, 2015; Keine et al., 2016). This non-monotonicity could render sound-driven firing rates close to spontaneous ones, but for most experimental stimuli tested, the acoustically-evoked SBC firing rates are still well above the spontaneous rate, including pure tones, amplitude-modulated, frequency-modulated and complex synthetic sounds (Kopp-Scheinpflug et al., 2002; Kuenzel et al., 2011, 2015; Keine and Rübsamen, 2015; Keine et al., 2016). To emphasize the finding of constant SBC output rates during natural stimulation, we now added an additional panel to Figure 1 (Figure 1Di) depicting the change in firing rates for both ANF input and SBC output during pure tone stimulation. We have added this information to the manuscript in the Results section when describing Figure 1.

How should we think about specificity in the output of SBCs vs. generality of the inhibitory input?

Considering the broad on-CF inhibition, we think that inhibitory inputs are recruited over a broad spectral range and integrated over a limited amount of time (the latter is captured in our current inhibition model) whenever the acoustic stimulus is close to the unit’s CF. We think that these two factors define the instantaneous strength of the inhibition. In our previous study, we also observed increased SBC output rates in response to various (complex) synthetic sounds. However, natural stimuli differ from the used synthetic stimuli both in their instantaneous spectral width, but also other spectrotemporal properties. We hypothesize that the inhibition integrates the properties, which then leads to a stronger gain control (keeping the SBC output rates constant) and (relatively) higher increases in sparsity and reproducibility. Whether this is an adaptation to natural statistics specifically, or just spectrally broad stimuli, or those with a particular autocorrelation is a question that remains to be addressed in future studies. We have added a whole section on this question in the Discussion.

4) Subtractive inhibition (Figure 2). It seems the goal here is to show how inhibition enhances timing precision among bushy cells with respect to auditory nerve fibers. This is an important idea and should be fleshed out more, where alternatives are clearly delineated and tested. The hypothesis seems to be that temporal precision is enhanced when inhibition tracks activity with some delay. This is reasonable based on the measured EPSP(fail) rates. How these translate into the parameters for the inhibitory model should be clearly depicted. But additional analyses are needed to conclude that the pattern of inhibition is responsible for enhanced temporal precision. For example, how is temporal precision influenced by uniformly deleted spikes (tonic inhibition), randomly deleted spikes, using an opposite pattern of spike deletion (heavier following sparse activity), activity-dependent decay of different depths (I(0)? S?) and timecourses (tau1)? Also, to make the approach clear, it would be helpful to show examples of original spike train, modeled inhibition, modified spike train, and sparsity/reproducibility before and after such manipulation. Please plot on a timescale that shows the differences to best advantage. What is the variance referred to in the Results section?

In the present study, we attempt to describe the increase in sparsity and reproducibility from the ANF input to the SBC output by a phenomenological model based on previous observations.

As suggested by the reviewer we compared the activity-dependent subtractive inhibition with three other models of inhibition (i) pure subtractive inhibition, i.e. removal of certain number of events, (ii) pure divisive inhibition, i.e. removal of certain fraction of events, (iii) inverted activity-dependent inhibition, i.e. stronger inhibition after low ANF activity and evaluated the results on failure fraction, sparsity and reproducibility. All parameters were chosen to best match the experimentally observed failure rates for each unit. A pure subtractive inhibition resulted in generally higher values of sparsity and reproducibility than observed in the experiments, which is likely caused by the “holes” in the PSTH for sections with low ANF activity. Divisive inhibition, i.e. scaling of the ANF response could be well-matched to the experimental failure fractions, but failed to increase sparsity and reproducibility. The inverted activity-dependent subtractive inhibition resulted in slightly increased sparsity and reproducibility, but failed to generate failure rates in the range of experimentally observed ones, particularly missing out on units with failure rates >50%. The activity-dependent subtractive inhibition showed the best match with the experimental data in failure rates, sparsity and reproducibility. While the increase in failure fraction and sparsity could be well modeled, some spread in reproducibility was observed in some cells, which might be caused by the sensitivity of this measure to spontaneously occurring spikes. While we believe that this simple model provides a fairly good match to the observed data, it is arguably not perfect. We now added a comparison of these different models to Figure 2 – supporting Figure 1.

The parameters I_o_, S, and O have now been described in more detail in the Methods section alongside the introduction of the model. The chosen values for the parameters are now explained in more detail and a schematic is provided in Figure 2—figure supplement 1 to aid the reader on how these values were derived.

5) Population vs. paired analysis. The approach taken here was to use the entire population of auditory nerve activity and the entire population of bushy cell activity to try to reconstruct the sound signal. Please explain the reasoning behind this approach. It seems to introduce some problems, the most obvious being frequency sensitivity of individual units. That is, one does not expect a single cell to be able to represent much of a sound, and a correction for this was confusingly mentioned in passing (subsection “Stimulus reconstruction”). The critical issue is how completely the individual bushy cell represents the stimulus compared to its inputs. Doing this comparison cell by cell would be more meaningful.

We acknowledge that the reasoning for the choice of analysis was a bit brief in the manuscript, following the length restrictions of the 'Research Advance' format. We agree that single cell analysis has a lot of merits in its own right: the distribution of cellular characteristics can be analyzed and differences between cells can be detected. This is the aim of most sensory processing studies, for example also in our previous study (Keine et al., 2016), where SBCs were characterized by their rate-level functions, spectrotemporal receptive fields, modulation transmission, phase locking, sparsity and reproducibility of response.

However, in the present manuscript, our aim was different: we wanted to quantify the representation of the stimulus on the population level, i.e. in particular focusing on the combined 'information' available in the population. Importantly, if multiple neurons have similar BFs, their responses can be used together to reconstruct the stimulus, which could for example be relevant if firing rate limitations allow feature of the stimulus only to encoded partially by each neuron (even in one frequency location).

As the reviewer points out, this overcomes the limitation a single cell representation has, namely a very limited 'field of view' in frequency, hence, the impossibility to represent the entire stimulus. If one performed single cell reconstruction, the result would be very limited, and the results obtained are already captured by more traditional methods of analysis, which quantify the temporal precision of response.

In essence the population analysis accounts for the well-documented strategy of population coding, employed throughout the mammalian nervous system. Population decoding, as performed here in an explicit manner, attempts to estimate the ensemble representation of the stimulus (thus mimicking what other neurons in the system could decode from it).

Clearly, the stimulus reconstruction performed here with a limited sample of cells only constitutes a coarse estimate (maybe a lower bound) on the representation by the entire population. We would, however, hypothesize that the present analysis remains meaningful, since we perform the reconstructions for all three ANF, SBC, and EPSP_fail_ components for paired synapses. Hence, differences should reflect those introduced specifically at the endbulb junction.

Regarding the description of this process in the Materials and methods section: "The stimulus was restricted to 200-4500 Hz, slightly extending the range of the characteristic frequencies (CFs, [1.18, 3.05] kHz) of the present sample (…)." This description only indicates that the stimulus, which in principle ranges up to 16kHz was limited for the reconstruction to the encoded frequencies by the sampled cells. As the reviewer points out, trying to reconstruct beyond this range, would be as futile as trying to reconstruct a wide range of frequencies from a single neuron.

We hope to have convinced the reviewer of the differences and utility of the present analysis. Reiterating that the single cell analysis is largely covered in the forward sense in the previous publication, and Figure 1 and Figure 2 in the present manuscript, we provide a shortened version of the above argument to motivate the analysis in the introduction, methods and results of the present manuscript.

6) The temporal enhancement would be expected based on the mechanisms (potassium conductances, inhibition), following observations by Joris et al. (and others, including work by the present authors) on the improvements in temporal representation of simple sounds in the bushy cell pathways. However, here the sharpened representation of temporal features of the stimulus likely does not derive from the ability of bushy cells to carry the timing of stimulus "fine structure", but it appears to be in carrying timing of envelope fluctuations, at least at the scale of analysis used here. Although this is interesting, it should be clarified, as the term "fine-structure" as used in the manuscript is not always consistent with its use in other parts of the literature, where it refers more closely to the instantaneous waveform than the waveform over a several msec period.

We agree, that our terminology was not appropriate and could lead to confusions with the existing literature. We now refer to it as “envelope fine-structure” to better distinguish it from the pure-tone phase-related fine-structure. The term has been replaced throughout the manuscript.

7) The discussion should touch more carefully on the improvement in sparsity and reproducibility for the natural stimuli compared to the synthetic stimuli used in the previous study, both in terms of potential mechanisms and the importance (if there is one) of the representation. How do the authors think this change came about, and why or how is it important?

We welcome the opportunity to add an interpretation of the respective findings. Overall, we hypothesize that the relatively higher increase in sparsity and reproducibility for natural stimuli is in fact relevant, although we would not claim that it would pertain solely to natural stimuli, but extend to other, synthetic stimuli, which match the natural stimuli in some of their properties. What these subsets precisely are, is an interesting question. In studying the cortex, researchers have previously attempted to combine natural speech stimuli with synthetic, general stimuli (SPORC = Speech + TORC, where TORCs are band-limited white-noise-type stimuli (David et al., 2009). There the purpose was to estimate neural tuning in the context of naturalistic temporal modulations. In the SBCs, further experiments, using similar kinds of hybrid stimuli would be required to tease apart what drives the increase sparsity and reproducibility. We predict that the high spectral density of natural stimuli is a main contributor to the level of increase (compare e.g. Figure 7A in Keine et al. 2016 with Figure 1 here), as the tuning of the inhibitory input is quite broad and will thus be driven strongly by broad-band stimulation. Another difference between artificial and natural stimuli is that the latter will to some degree contain correlations of second order (for textures, such as sand, but even higher for bird song), while the placement of the sounds in the RGS stimulus was chosen randomly. However, we have no evidence at this point to suspect that this would have an influence on sparsity and reproducibility. While potentially there could be in addition a dependence on the level of stimulation, this still is unlikely to have contributed here, as the stimuli were presented at the same average level. A briefer version of this argument has been added to the Discussion, combined with the response to Essential Revision #8 below, which also addressed the relation between natural and artificial stimuli.

8) The paper would benefit from discussion of other studies that have considered natural vs. artificial stimuli (e.g. Rieke, Bodnar and Bialek, (1995), Reinagel, (2001)). Finally, you should address how these observations relate to localization (or not) or other aspects of parsing and/or reconstructing the acoustic environment. Some concepts related to the framework of recent work from the last author's group (Boubenec et al., 2017) might present an interesting starting point.

We agree that a number of important thematic relationships were not appropriately addressed in the previous version, largely due to word limitations. Indeed, processing and encoding of natural sounds has been a recurring topic in (auditory) neuroscience, which has in fact often highlighted an apparent match between (certain) natural stimuli and the neural activity/system. Before, we only mentioned the classical findings by Lewicki, (2002), but we have now added related findings by Attias and Schreiner, (1997); Nelken et al., (1999); Hsu et al., (2004); Chechik and Nelken, (2012), reviewed in Theunissen and Elie, (2014) as well as your excellent suggestion of Rieke et al., (1995). We have also taken up your suggestion of interpreting the function of the inhibitory gain control from the perspective of predictive/contextual encoding and have dedicated the second-to-last section in the Discussion to this topic.